# Recurrent State Encoders for Efficient Neural Combinatorial Optimization

## Abstract

The primary paradigm in Neural Combinatorial Optimization (NCO) are construction methods, where a neural network is trained to sequentially add one solution component at a time until a complete solution is constructed. We observe that the typical changes to the state between two steps are small, since usually only the node that gets added to the solution is removed from the state. An efficient model should be able to reuse computation done in prior steps. To that end, we propose to train a recurrent encoder that computes the state embeddings not only based on the state but also the embeddings of the prior state. We show that the recurrent encoder can achieve equivalent or better performance than a non-recurrent encoder even if it consists of $3\times$ fewer layers, thus significantly improving on latency. We demonstrate our findings on three different problems: the Traveling Salesman Problem (TSP), the Capacitated Vehicle Routing Problem (CVRP), and the Orienteering Problem (OP) and integrate the models into a large neighborhood search algorithm, to showcase the practical relevance of our findings.

## 1 Introduction

Neural Combinatorial Optimization (NCO) is concerned with learning heuristics parameterized by deep neural networks for hard combinatorial optimization problems (COPs). The motivation is twofold: First, tailoring traditional heuristics to the exact problem at hand can be a difficult and time-consuming task, requiring specialized knowledge. If data-driven methods can be designed that are able to automatically learn high quality heuristics, development effort could be significantly reduced. Second, NCO uses the fact that for applications there always exists an implicit distribution over the instance space. Given the NP-hard nature of most problems, it is unlikely that there exists a single method that solves all instances efficiently. Thus, being able to specialize to any particular instance distribution by learning from data is a desirable property.

The primary paradigm currently are construction methods, where a neural network sequentially adds to a partial solution until some completion criterion is met. In the Traveling Salesman Problem (TSP), for example, the model would start at some node and iteratively add one of the not yet chosen nodes to the solution until all nodes are visited. Given that the goal is to solve optimization problems and with infinite time even naive enumeration of the solution space would yield the optimal solution, it is clear that finding good solutions quickly is the main goal. In the described construction process many problems inhibit the property that subsequent states are very similar. In the TSP for instance, every node that gets added to the solution, can be removed from the state, since it is already decided on and not relevant for future decision-making. This makes all pairs of subsequent states very similar, since they only differ in the one node.

Past approaches (Kool et al., 2019; Kwon et al., 2020; Xin et al., 2021a; Berto et al., 2024b) have therefore mainly relied on an encoder-decoder architecture, where the encoder computes a set of node embeddings only in the first step and the decoder computes the action probabilities based on these embeddings for all following steps. While efficient, it was observed that the models can struggle to learn with increasing problem size (Joshi et al., 2022) since the embeddings also contain information about increasingly less relevant interactions of components not present in the state anymore and recent work has shown that the more model capacity is added to the decoder instead of the encoder (Luo et al., 2023) or the more frequently the encoder is recomputed (Peng et al., 2020), the

better. In particular, Luo et al. (2023) only utilize a single encoder layer and Drakulic et al. (2023) remove the split between encoder and decoder entirely and apply a singular model at every step.

While this works well, it is significantly more expensive and ignores the similarity between the states. In order to still make use of this property and build a more efficient model, without losing the ability to adjust to changes in the state at every step, we instead propose to *learn the difference between subsequent states*. In order to do so, we train a recurrent encoder that computes the state embeddings not only based on the current state but also the embeddings of the step before, allowing it to reuse computation done in prior steps. We show that such an embedding update does not compromise on accuracy and decreases latency. Our contributions can be summarized as follows:

- We propose a **novel recurrent state encoder for neural combinatorial optimization**, which updates the node embeddings at every step based on the current state and the prior node embeddings. A hyperparameter $k$ controls the number of steps after which a non-recurrent base encoder is used to recompute the embeddings, allowing a flexible trade-off between both encoders.

- We demonstrate that the recurrent encoder can achieve equivalent or better performance to the non-recurrent encoder with significantly smaller number of parameters and thus improves on the latency accuracy trade-off. Depending on the exact model and problem, we find **latency decreases of** $1.8 - 4\times$ **at no significant accuracy drop**. Crucially, non-recurrent encoders of the same size and latency are not able to achieve the same performance. Additionally, the models are surprisingly robust, often delivering stable performance, even if the recurrent encoder is used with much larger $k$ than seen during training.

- Finally, we integrate our recurrent models into a large neighborhood search algorithm, showcasing how our improvements impact practically relevant search methods in terms of performance and latency. We demonstrate that our findings hold on three different combinatorial optimization problems: the Traveling Salesman Problem (TSP), the Capacitated Vehicle Routing Problem (CVRP), and the Orienteering Problem (OP).

## 2 RELATED WORK

**Neural Combinatorial Optimization** NCO has seen a diverse set of methodologies in recent years. In our work we focus on the very common constructive paradigm, where solutions are sequentially constructed with a neural network. Note however that various approaches exist, such as parameterizing a local search operator (Ma et al., 2021; 2023), learning a local search meta controller (Falkner et al., 2023; Xin et al., 2021b), parameterizing insertion operators (Hottung et al., 2025b; Khalil et al., 2017), learning heatmaps (Fu et al., 2021; Joshi et al., 2019; Min et al., 2023; Sun and Yang, 2023; Xin et al., 2021b; Li et al., 2023; Ye et al., 2024), learning to select subgraphs or decompositions (Falkner and Schmidt-Thieme, 2023; Hottung and Tierney, 2022; Li et al., 2021; Luo et al., 2023; Ye et al., 2024) and various hybridizations thereof.

In the domain of constructive methods our main contribution lies in the way the neural network processes the state. We propose a novel model that computes state embeddings from the current state and previous embeddings, thus only having to learn the difference between two states. In contrast, most prior work has focused on an encoder-decoder model (Kool et al., 2019) and variations of it (Berto et al., 2024b; Falkner and Schmidt-Thieme, 2020; Hottung et al., 2025a; Jin et al., 2023; Kwon et al.; Peng et al., 2020; Xin et al., 2021a), where the encoder computes a set of static node embeddings in the first step and the decoder computes the action probabilities based on these embeddings and some dynamic context information for all subsequent steps. It was shown however that such approaches struggle with increasing problem size (Joshi et al., 2022) and recent methods increasingly move capacity from the encoder to the decoder or frequently reembed the state (Peng et al., 2020; Xin et al., 2021a; Luo et al., 2023). At the extreme, when either moving all model capacity to the decoder or recomputing the encoder at every step, the split between encoder and decoder is removed entirely, which was shown by Drakulic et al. (2023) to perform much better.

Other work on constructive models focuses on either the training strategy or the search component. Various RL based training methods have been proposed (Kool et al., 2019; Kwon et al., 2020; Berto et al., 2024a), some auxiliary tasks (Kim et al., 2022) or curriculum strategies (Xin et al., 2021a) as well as recently self-improvement methods (Luo et al., 2023; 2024; Pirnay and Grimm, 2024a;b),

where the model searches for improved solutions during training, creating its own data. To improve the search, the literature has proposed tree search methods (Kwon et al., 2020; Choo et al., 2022; Pirnay and Grimm, 2024a), increasing solution diversity through multiple decoder heads (Xin et al., 2021a; Hottung et al., 2025a), as well as gradient based test time search, where some model parameters are adapted iteratively at inference time (Hottung et al., 2022; Choo et al., 2022; Hottung et al., 2025a). Since these aspects are not our focus, we stick to a simple beam search strategy and imitation learning for our models, but note that the mentioned literature could be integrated at a later time. Even without such advanced strategies, we find our models perform well.

Finally, we note that given the recent success of foundation models in language tasks, there is also a push in the NCO community for models that are trained on multiple combinatorial tasks (Drakulic et al., 2025; Berto et al., 2024b). In this work however, we stick to single task models.

**Recurrent Actors in RL**  Recurrent policies (or memory-based policies) (Hausknecht and Stone, 2015; Heess et al., 2015; Kapturowski et al., 2018; Morad et al., 2024) by themselves are not novel in Reinforcement Learning, with recent work also investigating sequence models, processing the entire state sequence with transformers and other sequence models (Bauer et al., 2023; Chen et al., 2021; Ni et al., 2023; Morad et al., 2024). However, their main application is in the context of partially observable environments. When the environment is not markov, the optimal action can depend on the entire state history and as such RNNs have been used to give the agent access to this history. Our environments however are markov. We instead make the observation that the step-by-step changes in the states are very small. A recurrent policy can reuse computation done in prior steps and only has to learn the differences between states. This enables more efficient models, which is especially important in combinatorial optimization where the policy is integrated into a larger search procedure and has to be evaluated many times.

**Speculative decoding**  We can also draw parallels to speculative decoding (Stern et al., 2018; Chen et al., 2023; Leviathan et al., 2023; Ankner et al., 2024) which aims to speed up inference of large autoregressive transformers, especially LLMs, by using smaller draft models to generate candidate continuations, which later get verified by the base model. Recent work uses a small recurrent head on top of the embeddings of the base model which bears similarity to our recurrent encoder (Ankner et al., 2024). However, there are some differences. Besides the obvious scale difference to LLMs, our tasks do not allow for causal attention. We have to recompute all pairwise interactions at every decoding step which is not the case in generative language modeling since tokens only attend to prior tokens. Additionally, we do not perform verification, which requires the base model to be computed for all steps even if some of them can be performed in parallel, enabling the speedup for speculative decoding methods. Since our decoding is always part of a larger search, we believe verification is not critical. Small accuracy drops can be compensated by searching more with the saved time.

## 3 METHOD

### 3.1 PROBLEM FORMULATION AND CONSTRUCTION PROCESS

We consider combinatorial optimization problems whose solutions can be sequentially constructed by iteratively adding variables from a discrete set to a partial solution until some completion criterion is reached. A COP instance $G \in \mathcal{G}$ consists of a finite set of feasible solutions $X_G$ and an objective function $f_G : X_G \to \mathbb{R}$. The goal is to find the optimal solution $x^* := \min_{x \in X_G} f_G(x)$.

In order to find solutions, we formulate a markov decision process (MDP) in which a policy, parameterized by a neural network, is used to sequentially construct a solution. Specifically, we utilize the recursive MDP formulations proposed in Drakulic et al. (2023) in which after every construction step the new state represents a reduced subproblem of the same problem class.

To illustrate, consider the well-known traveling salesman problem (TSP). Informally, the goal is, given a set of cities, to find the shortest cycle, that visits each city exactly once. Starting from any city, a solution can be constructed, by iteratively adding one of the not yet chosen cities, until all points are visited and a return to the starting city is made. To make the problem formulation recursive, it is redefined to find the shortest path instead of cycle for a set of points, given a starting and end point. This is referred to as the path-TSP problem. At every step, the newly chosen point

becomes the new starting point and the prior starting point is removed from the problem, such that at every construction step $t$ a valid path-TSP instance $G_t \in \mathcal{G}$ is presented to the policy. To recover the original TSP formulation, the starting point is duplicated and also added as the end point.

Besides the TSP, we also consider the Capacitated Vehicle Routing Problem (CVRP) and the Orienteering Problem (OP). Extended descriptions of the problems and their recursive formulations can be found in appendix A.

## 3.2 Model

Let $G_t \in \mathcal{G}$ be the remaining instance to be considered at time $t$, and $n_t$ the number of nodes in the instance. Our model consists of three components: a base encoder $E$, a recurrent encoder $U$, and a decoder $D$. The encoders need to produce embeddings $h_t \in \mathbb{R}^{n_t \times d_E}$ for $G_t$, where $d_E$ is the embedding dimension. The base encoder $E_{\theta_E}(G_t) \mapsto h_t$, parameterized by $\theta_E$, is a function that maps the instance $G_t$ directly to a set of node embeddings $h_t$, whereas the recurrent encoder $U_{\theta_U}(h_{t-1}, G_t) \mapsto h_t$, parameterized by $\theta_U$, is a function that updates the node embeddings based on the previous embeddings and the current state. As such it can reuse computation done in prior steps in order to be more efficient than the base encoder, especially when the step by step changes in the state are small. The decoder $D_{\theta_D}(h_t) \mapsto \Delta^{n_t}$, parameterized by $\theta_D$, is a function that maps the node embeddings to a probability distribution over the $n_t$ nodes.

**Base Model** In all our problems, the state $G_t$ is represented as a feature matrix $s_t \in \mathbb{R}^{n_t \times d_{\text{feat}}}$ of $n_t$ nodes each with $d_{\text{feat}}$ features. For the base encoder, we use a $L_E = 9$ layer transformer with ReZero (Bachlechner et al., 2021) connections and RMSNorm (Zhang and Sennrich, 2019) applied before the MHA and Feedforward blocks. The feed-forward networks are two-layer MLPs with ReLU activations, with model embedding dimension $d_E = 192$ and feed-forward dimension $d_{\text{FF}} = 512$. A node-wise linear layer is used to compute the initial node embeddings. The decoder is a single linear layer followed by a softmax, where infeasible actions are masked away by setting the logits to $-\inf$.

**Recurrent Model** Given the state representation $s_t \in \mathbb{R}^{n_t \times d_{\text{feat}}}$ at time $t$ and the previous embeddings $h_{t-1} \in \mathbb{R}^{n_{t-1} \times d_E}$, the recurrent encoder needs to compute the updated embeddings $h_t \in \mathbb{R}^{n_t \times d_E}$, from which the decoder produces the action distribution. Note that for all considered problems the prior step $s_{t-1}$ contained $n_t + 1$ nodes, since the previously selected node becomes the new starting node and the prior starting node is removed from the problem, since it is not relevant for the future decision-making anymore.

In order to align the prior embedding $h_{t-1}$ with the current state $s_t$, we remove the node embedding of the node that was removed from the problem in $s_t$ and call the resulting embeddings $\tilde{h}_t$. It is ensured, that the $i$-th element in $s_t$ and $\tilde{h}_t$ correspond to the same node.

An initial embedding for $s_t$ is then computed via a node-wise linear layer. Additionally, learnable start and learnable end-embeddings $h_{\text{start}}, h_{\text{end}} \in \mathbb{R}^{d_U}$ are added to the embedding of the current starting and final destination node, respectively. By convention, we order the nodes such that the start node is always the first node and the end node is always the last node. For notational convenience, we drop the time index $t$ in the following. The initial embeddings of node $i$ are then computed as

$$h_i^0 = \begin{cases} s_i W^0 + b^0 + h_{\text{start}} & \text{if } i = 1 \\ s_i W^0 + b^0 + h_{\text{end}} & \text{if } i = n \\ s_i W^0 + b^0 & \text{otherwise} \end{cases}, \tag{1}$$

where $W^0 \in \mathbb{R}^{d_{\text{feat}} \times d_U}$ and $b^0 \in \mathbb{R}^{d_U}$ are the learnable parameters. The prior embeddings $\tilde{h}$ are then combined with the current state embeddings $h^0$ as follows. First, an RMSNorm layer is applied to $\tilde{h}$, since these come from a possibly longer recurrent chain of repeatedly updated embeddings: $\hat{h} = \text{RMSNorm}(\tilde{h})$. Then the current and prior embedding are combined via concatenation and an MLP-layer with a residual connection, bringing $h_i^1$ to the same dimension as $h_i^0$

$$h_i^1 = \text{ReLU}([\hat{h}, h^0]_i W^1 + b^1) + h_i^0, \tag{2}$$

where $W^1 \in \mathbb{R}^{d_U + d_E \times d_U}$ and $b^1 \in \mathbb{R}^{d_U}$ are the learnable parameters. The resulting embeddings $h^1$ are then passed through $L_U$ blocks of multihead self-attention, normalization, and feedforward

networks to compute the updated embeddings. These blocks have the same structure as in the base model. The resulting embeddings are finally projected back to the original embedding dimension $d_E$ via a linear layer, and the result is used as the updated embeddings $h_t$, from which the decoder computes the action probabilities.

At inference time, the base encoder is used to compute the embeddings at step $t = 0$. From step $t = 1$ onwards, the recurrent encoder can be used to compute the embeddings. We include an optional hyperparameter $k$ that allows the base encoder to recompute the embeddings without the recurrence every $k$ steps. The procedure is illustrated in algorithm 2 and contrasted against only using the base model for inference without the recurrent encoder in algorithm 1. For further delineation from encoder-decoder models, see appendix B.

---

**Algorithm 1** Greedy inference with our base model

**Require:** $E_{\theta_E}, D_{\theta_D}, s_0$
  $t \leftarrow 0$
  done $\leftarrow$ False
  **while** not done **do**
    *// Compute logits and take the likeliest action*
    $a_t \leftarrow \mathrm{argmax}_a D_{\theta_D}(E_{\theta_E}(s_t))_a$
    $s_{t+1}, \text{done} \leftarrow \text{step}(s_t, a_t)$
    $t \leftarrow t + 1$
  **return** $a_0, ..., a_t$

---

**Algorithm 2** Greedy inference with our recurrent model

**Require:** $E_{\theta_E}, D_{\theta_D}, U_{\theta_U}, k, s_0$
  $t \leftarrow 0$
  done $\leftarrow$ False
  **while** not done **do**
    **if** $t \bmod k = 0$ **then**
      *// Initial or occasional reembedding*
      $h_t \leftarrow E_{\theta_E}(s_t)$
    **else**
      *// Recurrent update of embeddings from $h_{t-1}$*
      $h_t \leftarrow U_{\theta_U}(h_{t-1}, s_t)$
    $a_t \leftarrow \mathrm{argmax}_a D_{\theta_D}(h_t)_a$ *// Compute likeliest action*
    $s_{t+1}, \text{done} \leftarrow \text{step}(s_t, a_t)$
    $t \leftarrow t + 1$
  **return** $a_0, ..., a_t$

---

### 3.3 TRAINING

Since our main contribution is demonstrating the efficiency of the recurrent encoder, we train all models by imitation learning, following recent literature (Drakulic et al., 2023; Luo et al., 2023). While this is not optimal since the models will encounter distributional shifts through error accumulation at inference time, it eases the computational burden and additional complexity incurred by RL algorithms. Other training strategies may be used in future work to further improve the performance and other work have demonstrated that similarly sized models to our base model can be trained without labels by "self-improvement", where the models get used to search during the training process to iteratively create and improve their own data (Luo et al., 2024; Pirnay and Grimm, 2024a;b). More details can be found in appendix C.

### 3.4 LARGE NEIGHBORHOOD SEARCH

Since in many applications, it is unlikely that the model can reliably find the best solution by greedy construction, we integrate our recurrent model into a simple large neighborhood search (LNS) algorithm, to demonstrate the practical relevance of our findings. The LNS algorithm is a common metaheuristic that iteratively improves a solution by exploring a subset of the solution space. Our approach is described in algorithm 3. We use a beam search with the recurrent model to construct an initial solution $x$. Then at each iteration, we extract a subproblem based on the current solution and use the model to search for a better solution in the subproblem. If a better solution is found, we update the current solution by replacing the corresponding segment. The algorithm can be configured by the beam width used for the initial solution $b_{\text{init}}$ and the subproblems $b_{\text{sub}}$, how often to recompute the embeddings with the encoder $k_{\text{init}}, k_{\text{sub}}$ for both cases and the subproblem size $n_{\text{sub}}$.

In the TSP, we create multiple subproblems at each step, by extracting random non-overlapping segments of the current solution, each of size $n_{\text{sub}}$. The first and last nodes of each segment become the starting and end nodes of the path-TSP instance, and the order of the intermediate nodes can be reconsidered by the model. In the CVRP, since the model was trained only on instances where the

designated end node is also the depot, we only extract such segments. The first node however can be a customer node. Since due to this requirement, it is more cumbersome to extract multiple non overlapping segments, that fulfill this condition, we only extract a single segment of size $n_{sub}$. To do so, the solution $x$ is represented as a sequence. Since the order of the routes is arbitrary, we arrange their order uniformly at random at every step, increasing the diversity of subproblems.

---

**Algorithm 3** Large Neighborhood Search

---

**Require:** $G, U, D, k_{init}, k_{sub}, b_{init}, b_{sub}, t_{max}, n_{sub}$         *// Instance, Model, LNS hyperparameters*
    $x \leftarrow$ beam_search$(G, U, D, k_{init}, b_{init})$                *// Find initial solution via beam search*
    **for** $t = 1$ **to** $t_{max}$ **do**
        $G_{sub}, f_{G_{sub}}, x_{sub} \leftarrow$ sample_subproblem$(G, x, n_{sub})$      *// subproblem based on current sol*
        $x_{sub\_new} \leftarrow$ beam_search$(G_{sub}, U, D, k_{sub}, b_{sub})$          *// Find subproblem solution*
        **if** $f_{G_{sub}}(x_{sub}) > f_{G_{sub}}(x_{sub\_new})$ **then**
            $x \leftarrow$ update_solution$(x, x_{sub}, x_{sub\_new})$           *// Update solution*
    **return** $x$                                         *// Return final solution*

---

# 4 EXPERIMENTS

## 4.1 EXPERIMENTAL SETUP

We evaluate our models on three different combinatorial optimization problems: the Traveling Salesman Problem (TSP), the Capacitated Vehicle Routing Problem (CVRP), and the Orienteering Problem (OP). For each problem, we use a dataset of 1,000,000 trajectories collected by Concorde (David L. Applegate et al., 2003), PyVRP (Wouda et al., 2024) and EA4OP (Kobeaga et al., 2018) respectively for training. Additionally, datasets for validation and testing are created for each problem containing each 1000 instances. For each problem we train the models on problems of size 100 and evaluate them on problems of size 100, 200, 500 and 1000. For generation, we follow the established protocols in the literature. Details can be found in appendix D. We compare our methods on two metrics for each problem: the relative gap and the solution time. The relative gap gives the percentage difference in solution quality to a reference solution: $\frac{100}{f(x)}(f(\hat{x}) - f(x))$, where $f$ is the objective function, $x$ the reference solution and $\hat{x}$ the to be tested solution. For the reference solution, we use the solvers that also generated the training data. The solution time is the average time it takes to solve a single instance of the problem. All times are measured on a machine with an Nvidia A4000 16GB GPU and an AMD EPYC 7713P. For the baselines, we mostly focus on other constructive models. We compare to BQ (Drakulic et al., 2023) and LEHD (Luo et al., 2023) which are the most similar to us, also being trained with imitation learning. LEHD also includes a similar LNS scheme. We also compare to a variety of the encoder-decoder models and search procedures using them, including POMO (Kwon et al., 2020), EAS (Hottung et al., 2022), SGBS (Choo et al., 2022), and MDAM (Xin et al., 2021a). Finally, we compare to GLOP (Ye et al., 2024), which learns to hierarchically decompose the problem. All baseline results are obtained from the publically available implementations and pretrained checkpoints and were rerun on our datasets with our hardware, to make the results comparable.

## 4.2 RESULTS

**Comparing base and recurrent models** Figure 1 shows our main experiment, where we train recurrent models of different sizes in terms of number of layers, embedding dimension and number of heads on top of our largest base model and compare them to the base model, as well as additional non-recurrent models with the same size and structure as the recurrent models. We evaluate on the TSP, CVRP and OP, always with a beam search and measure the relative gap and solution time.

The **recurrent models reduce latency**, while maintaining or even improving the solution quality. Including the overhead of the environment and solution cost calculations, the measured speedup factor on 100-sized problems is between a factor of 1.8 and 2.8, depending on the model configuration and problem. The speedup factor increases with the problem size, since the overhead of the environment and computing the solution cost reduce. As such we observe a speedup of up to 3.3×

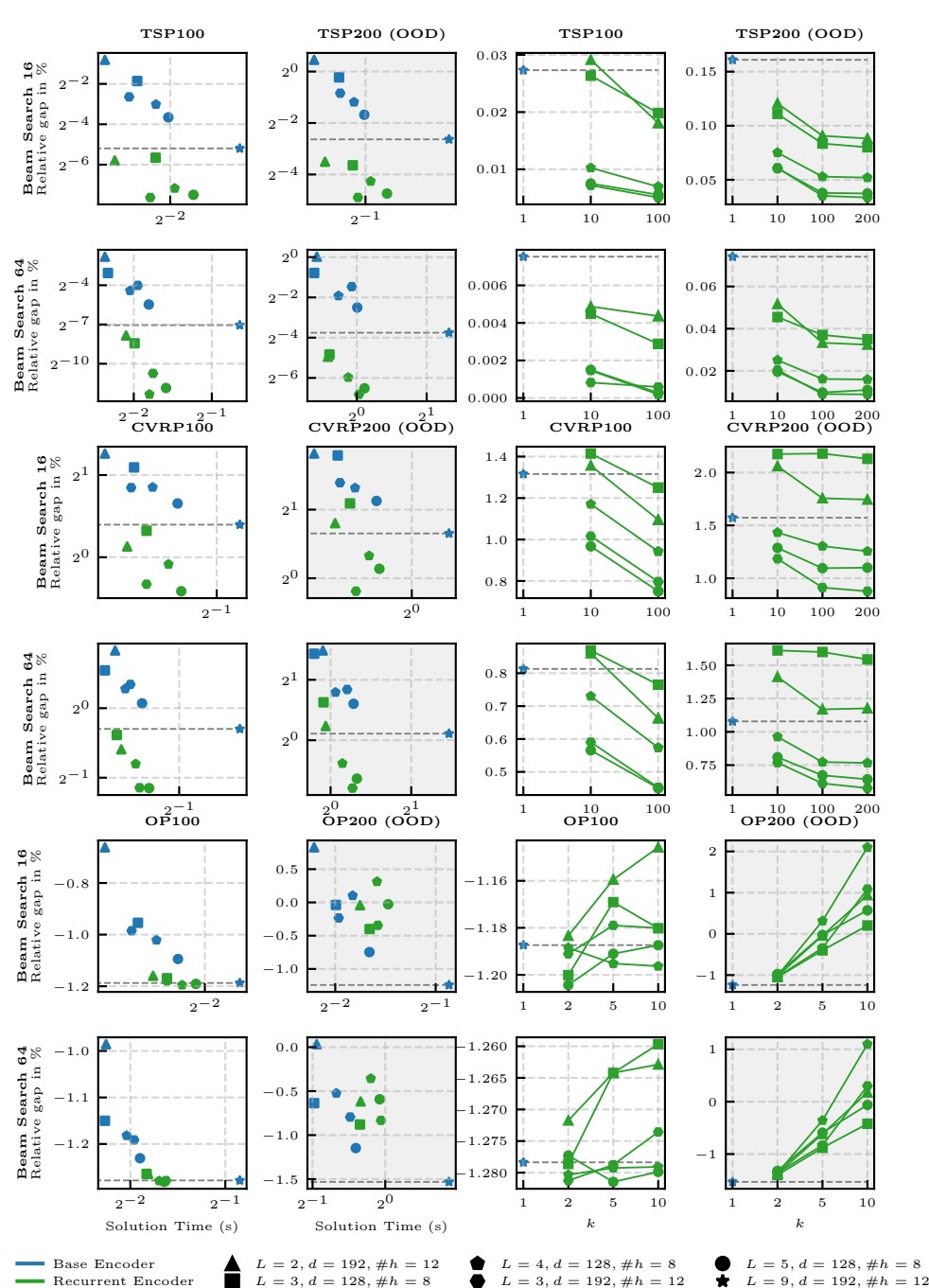

Figure 1: Main Results for recurrent models of different sizes vs base models of different sizes on the TSP, CVRP and OP. All models were trained on the same imitation learning dataset of 1 million trajectories with problems of size 100. Note that every point represents a trained model. Blue represents differently sized configurations of non-recurrent models, while green represents recurrent encoders having the respective same size and structure where $L$, is the number of layers, $d$ is the embedding dimension and $\#h$ the number of heads in the MHA mechanism. Recurrent models always use the largest base encoder (marked by ⭐) and are trained with $k = 10$. In the left two columns we show the relative gap of the models vs the time it takes to decode a single instance of the problem. The right two columns show the behavior of the models when the recurrent encoder is used with a larger $k$ than trained for. The possible ⭐ configuration is omitted since recurrent models should be smaller than their base model.

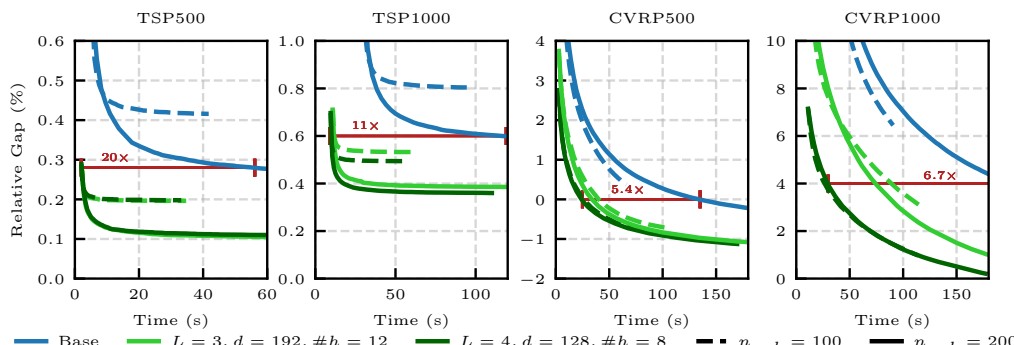

Figure 2: Results for Large Neighborhood Search with recurrent (green) and base (blue) models. All curves start when the initial solution has been constructed. The green curves start lower since the recurrent models found better initial solutions. Recurrent models are always used with $k_{\text{init}}, k_{\text{sub}} = 1000$, meaning we only ever use the base encoder for the first step. For the beam sizes, we use $b_{\text{init}}, b_{\text{sub}} = 16$ for the TSP and $b_{\text{init}} = 16, b_{\text{sub}} = 64$ for the CVRP, since in the CVRP, we only consider one subproblem at a time. We evaluate all models with two subproblem sizes $n_{\text{sub}} = 100, 200$. Note that none of the models are explicitly trained for the LNS setting.

on 200-sized problems and in appendix F we even observe a $4\times$ speedup on the TSP1000 with a beam size of 64 while still maintaining no accuracy drop relative to the base model.

Despite the significantly fewer active parameters and the reduced latency, **the recurrent models match or even exceed the performance of the base model** and especially their non-recurrent counterparts with the same size. Note that outperforming the base model is only possible since the recurrent models are also trained to predict the optimal action from the imitation learning dataset. Another option is to train the recurrent models to match the base models embeddings or action distribution, but this caps the best obtainable performance to that base model. This result shows that the recurrent models are effective at reusing computation from prior steps and do not simply ignore the previous embeddings but use them effectively to solve the task. This still holds true in the out of distribution (ood) settings with double the nodes. Only in the ood OP instances are the recurrent models not quite able to match the performance of the base model. However, on the OP all models report negative gaps to the solver that produced the training data, so while the recurrent models still might fit the training data better, given the inherent limitations of imitation learning, especially with suboptimal labels, accuracy might be reduced due too mimicking the solver too well.

Additionally, the **recurrent encoders are robust to the number of steps** $k$ that the recurrent encoder is used, before the base encoder is run again. While all models were trained only with $k = 10$, on the TSP and CVRP, the performance of the recurrent models actually increases with larger $k$ than seen during training, even in the ood settings and using the recurrent model with $k = 200$, a $20\times$ increase. The performance increase on the TSP and CVRP is due to recurrent models being better than the base models, thus running them for more steps is beneficial. Still, the stability of the recurrence even on ood instances is a nontrivial finding. Only on the OP, the results are more mixed, with stable performance in the in distribution setting but falling behind in the ood case.

**Application to LNS**  In Figure 2 we show results with LNS on larger problems of size 500 and 1000. As a trade-off between solution quality and time, we evaluate two intermediate configurations of our recurrent encoders with (i) $L_U = 3, d_U = 192, d_{\text{FF}} = 512, \#h = 12$ and (ii) $L_U = 4, d_U = 128, d_{\text{FF}} = 256, \#h = 8$ and compare them to only using the base model. While none of the models were explicitly trained for the larger problem sizes, or their subgraph distributions, we observe good performance. Additionally, we observe that the recurrent models clearly outperform the base model in terms of the trade-off between solution quality and time. In Figure 3 we expand on the results, adding all problem sizes and compare the results to the baselines. We can clearly see that our recurrent encoder with LNS outperforms all other methods in terms of the time-quality trade-off. To

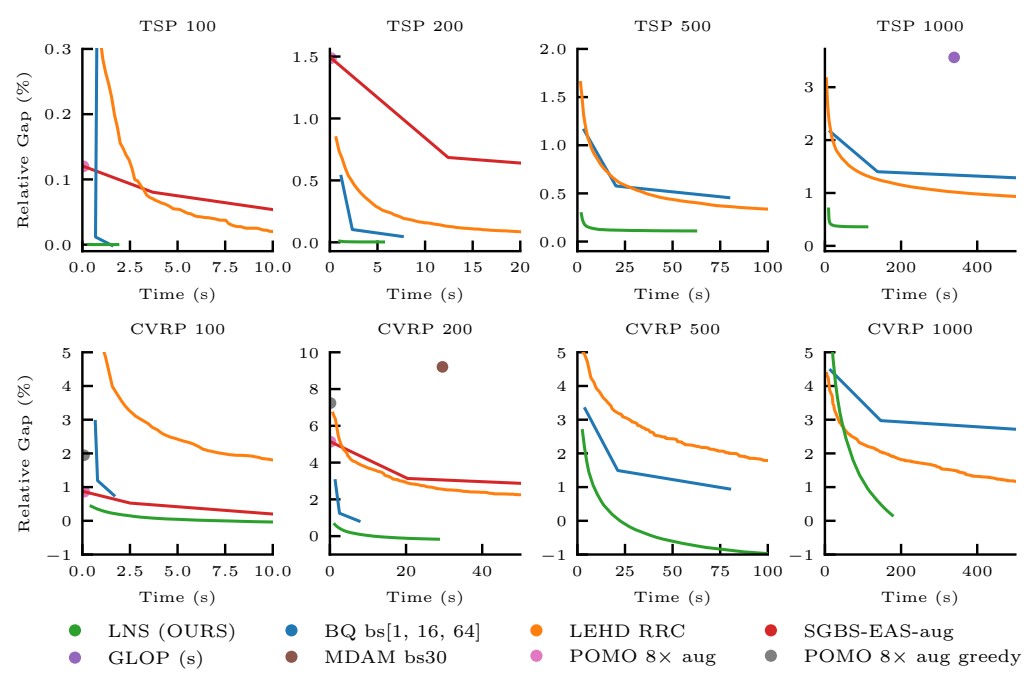

Figure 3: Comparison with baseline models on TSP and CVRP. For more information on the other methods, refer to section 4.1 and E. We limit the axes ranges for better visibility. Methods that are not visible in the plot perform outside the range (significantly worse), thus are not shown.

ensure that our recurrent models are the main contributor of the good performance of the LNS, we compare them directly to other models on typical subgraph sizes in appendix F.

## 5 CONCLUSION AND LIMITATIONS

We proposed a recurrent encoder for combinatorial optimization, enabling reuse of computation done in previous steps by updating the node embeddings based on the previous embeddings and current state. Thus, the model only needs to learn the difference between subsequent states. We demonstrated on the TSP, CVRP and OP that more efficient decision-making can be modeled, leading to latency decreases while increasing solution quality, especially in a large neighborhood setting.

As discussed in sections 3.3 and 2, the models were trained by imitation learning for computational efficiency and simplicity, since our main contribution lies in the recurrent encoder and not training strategies for NCO. RL or self-improvement training can be adopted in the future. We also note that while the recurrent encoder was trained in a separate training stage from the base model, it is possible to train both models jointly, potentially making the base encoder produce more "updatable" embeddings at the expense of increased training cost. If the field of NCO moves towards recent trends of large foundation models, our proposed two-stage training might fit the use case of having a large-scale pretrained model serve as the base model and then a much smaller recurrent encoder on top of it. In such a case the recurrent model could also be trained by imitating the base models action distribution, similar to knowledge distillation, instead of expert trajectories. Given the outlined options in the design space and their possible trade-offs, we believe our work opens up future work.

Lastly, we have demonstrated that our approach can work on the presented problems. However, likely there also exist problem types, where subsequent states in high quality solutions are not similar enough for efficiency gains. In the future, it needs to be further explored what larger problem classes are suitable for our modeling approach and if there exist further conditions such as a minimum quality and smoothness in the embeddings that the base model needs to fulfill.

**Reproducibility Statement**   To ensure reproducibility, we provide the source code, our pretrained models, our collected training datasets as well as test datasets at an anonymized repository[1]. Example commands for training and testing models are provided. Details about our model structure can be found in section 3.2 and 4.2. Training Details are found in 3.3 and C.

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

# A    PROBLEMS DETAILS

## A.1    TRAVELING SALESMAN PROBLEM

A TSP instance consists of a set of cities $C = \{1, \ldots, n\}$ and the associated pairwise distances $d_{ij} \in \mathbb{R}, i, j \in C$. A solution $x$ is a permutation of the cities $C$, such that $x_i \in C$ is the $i$-th city in the tour. The objective function is given via

$$f(x) \coloneqq \sum_{i=1}^{n-1} d_{x_i, x_{i+1}} + d_{x_n, x_1}$$

The model input at each step consists of the state $s_t \in \mathbb{R}^{n_t \times 2}$, where $n_t$ are the number of nodes and each node $s_{t,i} \in \mathbb{R}^2$ has two features, its 2D coordinates. The first node $s_{t,0}$ always represents the current position, while the last node $s_{t,n_t}$ always represents the destination node, that completes the cycle. As such in the first step, the starting node is duplicated and also added as the destination node, such that the objective of finding the shortest path between starting and destination node corresponds to the actual objective of finding the shortest cycle. The models output at every step is a probability distribution over the $n_t$ nodes, where infeasible actions are masked away.

## A.2    CAPACITATED VEHICLE ROUTING PROBLEM

A CVRP instance consists of a set of nodes $C = \{0, 1, \ldots, n\}$, where node 0 is the depot and the remaining nodes are called the customers. Each customer $i \in C \smallsetminus \{0\}$ has a demand $d_i \in \mathbb{R}$ and all nodes have associated pairwise distances $d_{ij} \in \mathbb{R}, i, j \in C$. A vehicle with capacity $Q$ must now serve all customers. As such, a feasible solution $x = \{r_1, \ldots r_{|x|}\}$ consists of a set of routes. Each route $r = \{0, \ldots, 0\}$ starts and ends at the depot and must serve some subset of customers, such that the cumulative demand of these customers does not exceed the vehicle capacity $Q$ and over all routes each customer is visited exactly once. The goal is to find such a solution $x$ that minimizes the total distance traveled, given by the objective function

$$f(x) \coloneqq \sum_{i=1}^{|x|} \sum_{j=1}^{|r_i|-1} d_{(r_{i,j}), (r_{i,j+1})}$$

where $|x|$ is the number of routes in the solution and $|r_i|$ is the number of nodes in route $i$, including the depots at the start and end of the route.

The model inputs at each step are similar to the TSP model. In addition to the 2D-coordinates, the state $s_t \in \mathbb{R}^{n_t+1 \times 4}$ contains the demand of each node, normalized by the vehicle capacity $Q$, as well as the remaining normalized capacity of the vehicle at the current step $t$. The first node $s_{t,0}$ always represents the current position, while the last node $s_{t,n_t}$ always represents the depot node, which also functions as the destination node. At the first step, again starting and end node are the same, and as such they are duplicated in the state. We follow (Drakulic et al., 2023) and instead of letting the model predict a distribution over the $n_t + 1$ nodes, including the depot, as the action, the model predicts a distribution over $2n_t$ actions, where for each customer, the model can select to either visit it directly or visit it via the depot. Infeasible actions are masked.

## A.3    ORIENTEERING PROBLEM

We consider a distance constrained version of the OP. An instance consists of a set of nodes $C = \{0, 1, \ldots, n\}$, where 0 is the depot node and all other nodes $i \in C \smallsetminus \{0\}$ are assigned a prize $p_i \in \mathbb{R}$. As in the other problems, the nodes have associated pairwise distances $d_{ij} \in \mathbb{R}, i, j \in C$. A feasible solution $x$ consists again of a sequence of customer visits, where each customer cannot be visited more than once, however in contrast to the other problems not all customers need to be visited, but there exists a total distance constraint $D$, which $x$ cannot surpass. The goal is to find a path starting and ending at the depot, such that the total prize of the visited nodes is maximized. As such, the objective is given by

$$f(x) \coloneqq \sum_{i=1}^{|x|} p_{x_i}$$

The model input is similar to the TSP. The only difference is that in addition to the 2D-coordinates, the prizes of each node normalized by the maximum prize and the remaining distance limit are added. The prize of the depot is always set to $0$.

# B    COMPARISON WITH ENCODER-DECODER MODELS

We have already contrasted our approach to policies that process all states separately, thus having no real encoder-decoder structure in section 3.2. This is the case for our base model or models like BQ Drakulic et al. (2023). However, the literature has early on already tried to make use of the fact all nodes of the problem are known at the start of the solution construction. As such, encoder-decoder models have been proposed Hottung et al. (2025a); Jin et al. (2023); Kool et al. (2019); Kwon et al.; 2020); Luo et al. (2023); Xin et al. (2021a). They typically only embed all nodes initially once and then let the decoder predict from these **static** embeddings, without updating them. To get information about the current state, and all occurred changes, a context node is provided with engineered features. While faster, this can be limiting, since the decoder has to operate from increasingly out of date embeddings. Our recurrent model has the advantage that it can update the embeddings to reflect the current state, while still reusing computation from prior steps. When step-by-step changes are small, this is more efficient and computation is divided more equally across the steps. To illustrate the difference again, we provide pseudocode for greedy inference with a typical encoder-decoder model and our recurrent model in algorithms 4 and 5. We also compare our recurrent models experimentally again to both other paradigms: (i) recomputing embeddings from scratch at every step and (ii) encoder-decoder models with static embeddings in appendix F.

---

**Algorithm 4** Greedy inference with a typical encoder-decoder model

**Require:** $E_{\theta_E}, D_{\theta_D}, s_0$
  $t \leftarrow 0$
  done $\leftarrow$ False
  $h_0 \leftarrow E_{\theta_E}(s_0)$    *// Compute embeddings only once at the start*
  **while** not done **do**
      *// Compute logits and take likeliest action with **fixed** embeddings $h_0$ and context from current state $s_t$*
      $a_t \leftarrow \mathrm{argmax}_a D_{\theta_D}(h_0, s_t)_a$
      $s_{t+1},$ done $\leftarrow$ step$(s_t, a_t)$
      $t \leftarrow t + 1$
  **return** $a_0, ..., a_t$

---

**Algorithm 5** Greedy inference with our recurrent model

**Require:** $E_{\theta_E}, D_{\theta_D}, U_{\theta_U}, k, s_0$
  $t \leftarrow 0$
  done $\leftarrow$ False
  **while** not done **do**
      **if** $t \bmod k = 0$ **then**
          *// Initial or occasional reembedding*
          $h_t \leftarrow E_{\theta_E}(s_t)$
      **else**
          *// Recurrent update of embeddings from prior step*
          $h_t \leftarrow U_{\theta_U}(h_{t-1}, s_t)$
      *// Compute logits and take likeliest action*
      $a_t \leftarrow \mathrm{argmax}_a D_{\theta_D}(h_t)_a$
      $s_{t+1},$ done $\leftarrow$ step$(s_t, a_t)$
      $t \leftarrow t + 1$
  **return** $a_0, ..., a_t$

---

# C    TRAINING AND HARDWARE DETAILS

For the imitation learning procedure, we first train the base model parameters $\theta_E, \theta_D$, and in a second training stage we train the recurrent model parameters $\theta_U$, while freezing the base model. Training for the base model is conducted by sampling a batch of expert trajectories from the dataset $\mathcal{D}$. Subsequently, a random step is chosen from the trajectory and the corresponding pair of state and ground-truth action is extracted from the trajectory. The model is then updated via the cross-entropy loss between the ground-truth action and the predicted action. For the recurrent encoder, we take the trained base model and again sample a batch of expert trajectories from the dataset $\mathcal{D}$. We again sample a random starting state, but now accumulate the cross-entropy loss over the next $k$ steps, where the embeddings are updated recurrently and the ground-truth trajectory is followed. The training procedure is illustrated in algorithm 6.

For all problem types, we train the models with the same set of hyperparameters, except the model structure, where number of layers, the hidden dimension, and feed forward dimension, and number

---

**Algorithm 6** Imitation Learning - Training for base and update model components

---

**Require:** $\mathcal{D}, E, U, D, k, M, \alpha$      // *Dataset, model components and hyperparameters*
  **for** $m = 1$ **to** $M$ **do**      // *Training of the base model*
    $(s_0, a_0, ..., s_T, a_T) \sim \mathcal{D}$      // *Sample expert trajectory from the dataset*
    $j \sim \mathrm{Uniform}(0, T)$      // *Sample a random step*
    $l \leftarrow \mathrm{CE}(a_j, D_{\theta_D}(E_{\theta_E}(s_j)))$      // *Compute cross-entropy loss*
    $(\theta_E, \theta_D) \leftarrow (\theta_E, \theta_D) - \alpha \nabla_{(\theta_E, \theta_D)} l$      // *Update base model parameters*
  **for** $m = 1$ **to** $M$ **do**      // *Training of the recurrent encoder*
    $(s_0, a_0, ..., s_T, a_T) \sim \mathcal{D}$      // *Sample expert trajectory from the dataset*
    $j \sim \mathrm{Uniform}(0, T - k)$      // *Sample a random starting step*
    $l \leftarrow 0$
    $h_j \leftarrow E_{\theta_E}(s_j)$
    **for** $i = j + 1$ **to** $j + k$ **do**
      $h_i \leftarrow U_{\theta_U}(h_{i-1}, s_i)$      // *Update embeddings recurrently*
      $l \leftarrow l + \mathrm{CE}(a_i, D_{\theta_D}(h_i))$
    $\theta_U \leftarrow \theta_U - \alpha \nabla_{\theta_U} l$      // *Update only $U_{\theta_U}$ parameters*
  **return** $\theta_E, \theta_D, \theta_U$

---

of heads are varied, as indicated in the figures. All models use $k = 10$ for training, as a trade-off between training time and performance, since the larger the $k$, the longer the sequence of steps that Backpropagation through Time (BPTT) needs to be computed for. We used the schedulefree Adam variant (Defazio et al., 2024) with a learning rate of $1e - 3$, training for a maximum of 1000 epochs except for the base model which was trained for 1500 epochs at maximum. We use early stopping with a patience of 100 epochs and always save the overall best model by greedy performance on the validation set.

## D  DATA GENERATION

For all datasets, we sample 2D coordinates uniformly at random from the unit square, independently for each node. The distance matrix is then computed via pairwise euclidean distance. We consider a problem of size $n$, to be a TSP, CVRP or OP instance with $n$ customer nodes, meaning for CVRP and OP, there is an additional depot node, making the problem contain a total of $n + 1$ nodes. In the CVRP, we sample the demand of each customer uniformly at random from $[1, 10]$, which is widely used, starting with (Kool et al., 2019). The vehicle capacity is dependent on the problem size and following the literature (Berto et al., 2024a;b) we choose $Q_{100} = 50$, $Q_{200} = 70$, $Q_{500} = 130$, $Q_{1000} = 230$, where $Q_n$ is the vehicle capacity for a problem of size $n$. For the OP, we follow Drakulic et al. (2023) and fix the distance constraint to $4$. The prize $p_i$ of customer $i$ is determined by its distance to the depot relative to maximum distance between the depot and any customer, making farther away customers more valuable:

$$p_i = 1 + \lfloor 99 \frac{d_{0i}}{\max_j d_{0j}} \rfloor$$

where $\lfloor \cdot \rfloor$ is the floor function. The score is thus between $[1, 100]$.

## E  BASELINE DESCRIPTIONS

**Concorde**   Concorde (David L. Applegate et al., 2003) is a widely known exact TSP solver. We used it as our reference solver for obtaining the training data for the TSP, as well as the reference solutions for all validation and test datasets. The code is freely available for academic purposes.

**EA4OP**   EA4OP (Kobeaga et al., 2018) is a metaheuristic, combining an evolutionary algorithm with a local search for the Orienteering Problem. We used it as our reference method for obtaining the training data for the Orienteering Problem, as well as the reference solutions for all validation

and test datasets. The implementation available in OPSolver [2] was used, which also provides an alternative exact solver. The code is freely available for academic purposes.

**PyVRP**   PyVRP (Wouda et al., 2024) is a meta heuristic framework for various routing problems, based upon the hybrid genetic search algorithm for the CVRP by Vidal et al. (2012); Vidal (2021). To generate the CVRP datasets, we used RL4CO (Berto et al., 2024a), which provides a built-in interface to PyVRP in its MTVRP implementation. The code is freely available for academic purposes.

**MDAM**   The Multi-Decoder Attention Model (Xin et al., 2021a) is an extension on the encoder-decoder style of neural combinatorial models (Kool et al., 2019), trained via reinforcement learning. MDAM changes the split between encoder and decoder and additionally adds multiple decoder heads. To promote diversity, the heads are regularized with a KL-Divergence loss to promote diversity among the multiple decoders. The code is available under the MIT license.

**POMO**   Policy Optimization with Multiple Optima (Kwon et al., 2020) is based on the encoder-decoder model by Kool et al. (2019) and proposed an improved training procedure and inference mechanism which forces the model to start from all possible starting nodes, which is especially helpful for the TSP, where the starting node is arbitrary. At inference time, the model additionally creates diverse solutions by creating multiple augmentations of the instance by rotating the coordinates. The code is available under the MIT license.

**EAS**   Efficient Active Search (Hottung et al., 2022) explores multiple variants of an active search approach, where the constructive model is updated by the Reinforcement Learning method during the inference procedure for the specific instance to be solved. Multiple variants are proposed: (i) only the embeddings are updated, (ii) an additional adapter layer is added to the decoder, (iii) a table is initialized that directly updates the logits. The source code is freely available.

**SGBS**   Simulation Guided Beam Search (Choo et al., 2022) is a tree search procedure inspired by Monte Carlo Tree search, which instead of only relying on the model probabilities, scores intermediate nodes by greedy rollouts of the policy, to subsequently adjust where to search next. It is additionally combined with the active search approach of Hottung et al. (2022). The code is available under the MIT license.

**BQ**   BQ (Drakulic et al., 2023) reframes the MDP, removing all already decided nodes from the state and action space. Consequently, the model is not split into an encoder and decoder, but instead a single deep network is computed at each step. The model is trained via imitation learning, where the training data is generated by a solver. The code is available under the CC BY-NC-SA 4.0 license.

**LEHD**   LEHD (Luo et al., 2023) is similar to BQ, being trained by imitation learning from solver generated solutions. The model still has a split between encoder and decoder, however all except one layer is placed in the decoder, in contrast to other encoder-decoder approaches. They additionally add a large neighborhood search which selects a subproblem by selecting a random subsegment based on the current solution and then resolving it with the model by greedy construction. The code is available under the MIT license.

**GLOP**   GLOP (Ye et al., 2024) focuses on large instances by hierarchically decomposing the problem into TSPs and those into path-TSPs. The decomposition is sampled from a GNN generated heatmap and the TSPs are solved by iteratively decomposing and solving segments with an encoder-decoder model. The code is available under the MIT license.

## F   ADDITIONAL RESULTS

In this section, we show additional results for the TSP and CVRP. First we demonstrate that the good performance of our Large Neighborhood search is tied to our better recurrent models by comparing

---

[2]https://github.com/gkobeaga/op-solver

Table 1: Detailed Comparison of our recurrent models without Large Neighborhood Search to BQ and LEHD models. We can conclude that also without the search our models demonstrate superior performance both in terms of absolute cost and runtime. Thus, replacing our model with others in our LNS procedure would not be helpful. All experiments are conducted on the same Nvidia A4000.

### Traveling Salesman Problem (TSP)

| Method | Variant | TSP100 | | | TSP200 (ood) | | |
|---|---|---|---|---|---|---|---|
| | | Cost | Gap | Time (s) | Cost | Gap | Time (s) |
| Concorde | | 7.768 | 0% | - | 10.697 | 0% | - |
| BQ | Greedy | 7.798 | 0.39% | 0.78 | 10.754 | 0.53% | 1.18 |
| | Bs16 | 7.769 | 0.01% | 0.69 | 10.708 | 0.10% | 2.36 |
| | Bs64 | 7.768 | 0.00% | 1.55 | 10.702 | 0.05% | 7.61 |
| LEHD | greedy | 7.810 | 0.54% | 0.31 | 10.787 | 0.84% | 0.65 |
| | rrc10 | 7.782 | 0.18% | 1.84 | 10.735 | 0.35% | 3.89 |
| | rrc100 | 7.769 | 0.01% | 15 | 10.704 | 0.06% | 32 |
| | rrc1000 | 7.768 | 0.00% | 154 | 10.699 | 0.02% | 324 |
| Ours l=4 k=200 | Greedy | 7.791 | 0.30% | 0.24 | 10.745 | 0.45% | 0.47 |
| Ours l=4 k=200 | Bs16 | 7.769 | 0.01% | 0.26 | 10.703 | 0.05% | 0.52 |
| Ours l=4 k=200 | Bs64 | 7.768 | 0.00% | 0.29 | 10.699 | 0.02% | 0.92 |
| Ours l=5 k=200 | Greedy | 7.793 | 0.32% | 0.26 | 10.733 | 0.34% | 0.53 |
| Ours l=5 k=200 | Bs16 | 7.769 | 0.01% | 0.29 | 10.701 | 0.04% | 0.58 |
| Ours l=5 k=200 | Bs64 | 7.768 | 0.00% | 0.33 | 10.698 | 0.01% | 1.08 |

### Capacitated Vehicle Routing Problem (CVRP)

| Method | Variant | CVRP100 | | | CVRP200 (ood) | | |
|---|---|---|---|---|---|---|---|
| | | Cost | Gap | Time (s) | Cost | Gap | Time (s) |
| PyVRP | | 15.5781 | 0% | - | 22.046 | 0% | - |
| BQ | Greedy | 16.037 | 2.95% | 0.68 | 22.711 | 3.02% | 1.40 |
| | Bs16 | 15.764 | 1.19% | 0.80 | 22.321 | 1.25% | 2.52 |
| | Bs64 | 15.696 | 0.76% | 1.66 | 22.227 | 0.82% | 7.66 |
| LEHD | greedy | 16.500 | 5.92% | 0.39 | 23.521 | 6.69% | 0.81 |
| | rrc10 | 16.089 | 3.28% | 2.46 | 23.012 | 4.38% | 4.66 |
| | rrc100 | 15.790 | 1.36% | 21.75 | 22.571 | 2.38% | 40.67 |
| | rrc1000 | 15.709 | 0.84% | 200.21 | 22.363 | 1.44% | 409.30 |
| Ours l=4 k=200 | Greedy | 15.964 | 2.48% | 0.35 | 22.710 | 3.01% | 0.69 |
| Ours l=4 k=200 | Bs16 | 15.722 | 0.92% | 0.36 | 22.317 | 1.23% | 0.79 |
| Ours l=4 k=200 | Bs64 | 15.664 | 0.55% | 0.38 | 22.211 | 0.75% | 1.11 |
| Ours l=5 k=200 | Greedy | 15.905 | 2.10% | 0.38 | 22.681 | 2.88% | 0.77 |
| Ours l=5 k=200 | Bs16 | 15.692 | 0.73% | 0.39 | 22.285 | 1.08% | 0.84 |
| Ours l=5 k=200 | Bs64 | 15.645 | 0.43% | 0.41 | 22.184 | 0.62% | 1.25 |

them to the BQ (Drakulic et al., 2023) and LEHD (Luo et al., 2023) models with greedy and beam search decoding on typical subproblem sizes. Both represent state-of-the-art models from the two alternative approaches of (i) having no split between encoder and decoder and just a single model (BQ) and (ii) a model from the encoder-decoder family (LEHD). The results are displayed in Table 1. We show that our models perform better in terms of absolute cost and runtime, thus replacing our models in our LNS procedure with others would not be helpful.

Additionally, we present extended results of the main figure 1 to ood problem sizes of up to 1000. On the CVRP, not only the problem size is ood, but also the vehicle capacity increases, which dramatically changes the typical length of the individual routes. As discussed in section D, the vehicle capacity for the training instances was set to $Q_{100} = 50$. For the largest instances here, it is set to $Q_{1000} = 230$. We can see that the recurrent models still perform quite well on most ood cases. Especially on the TSP, performance is remarkably consistent with the largest recurrent model, delivering $\approx 0.5\%$ gap to the optimal concorde solutions on the 1000-sized instances with a beam search, despite those instances being $10\times$ larger than the training instances and the recurrent encoder being used a $100\times$ more steps in a row without recomputing the embeddings ($k = 10$ vs $k = 1000$).

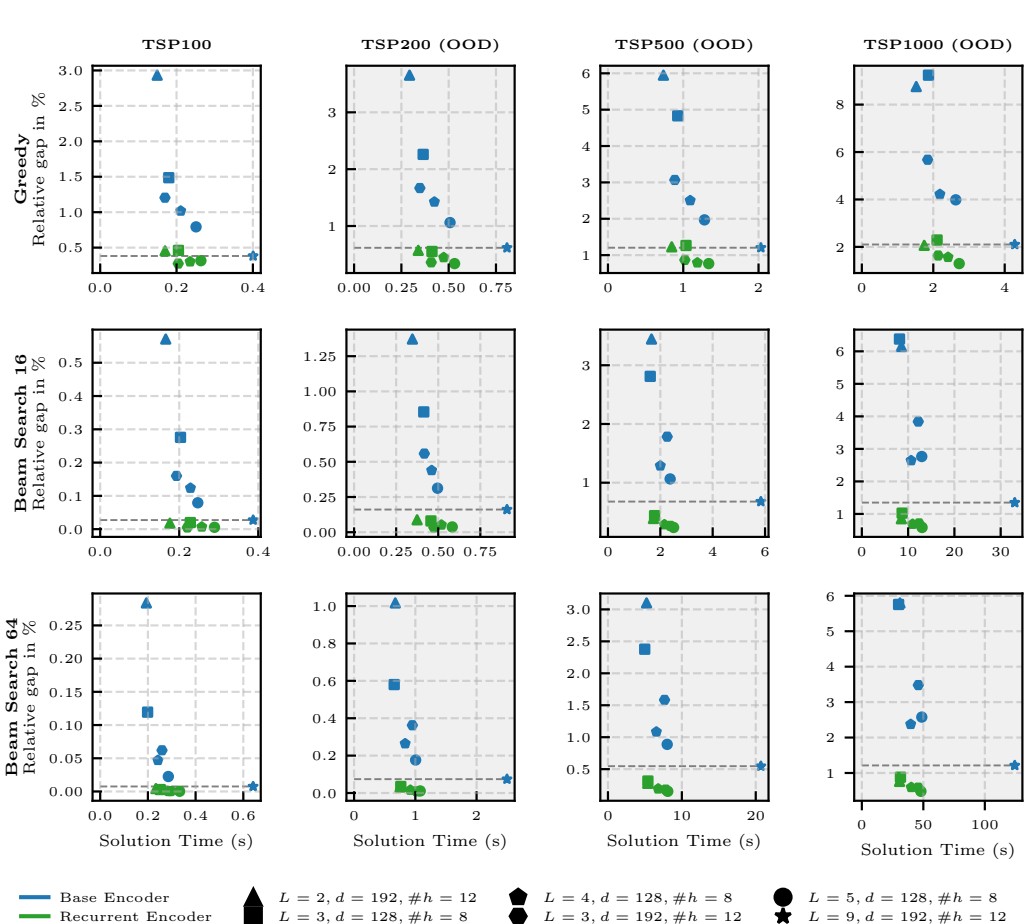

Figure 4: Additional results for recurrent models of different sizes vs base models of different sizes on the TSP. All models are the same as in the main paper (see Fig 1). We show the relative gap of the models vs the time it takes to decode a single instance of the problem on ood instances of up to size 1000. All models use maximum $k = 1000$. For results with varying $k$, see figure 5. All models were trained on the same imitation learning dataset of 1 million trajectories with problems of size 100. The models in blue are differently sized configurations of non-recurrent models, while the models in green are recurrent with the recurrent encoder having the respective same size and structure where $L$, is the number of layers, $d$ is the embedding dimension and $\#h$ the number of heads in the MHA mechanism. All recurrent models always use the largest available base encoder and are trained with $k = 10$.

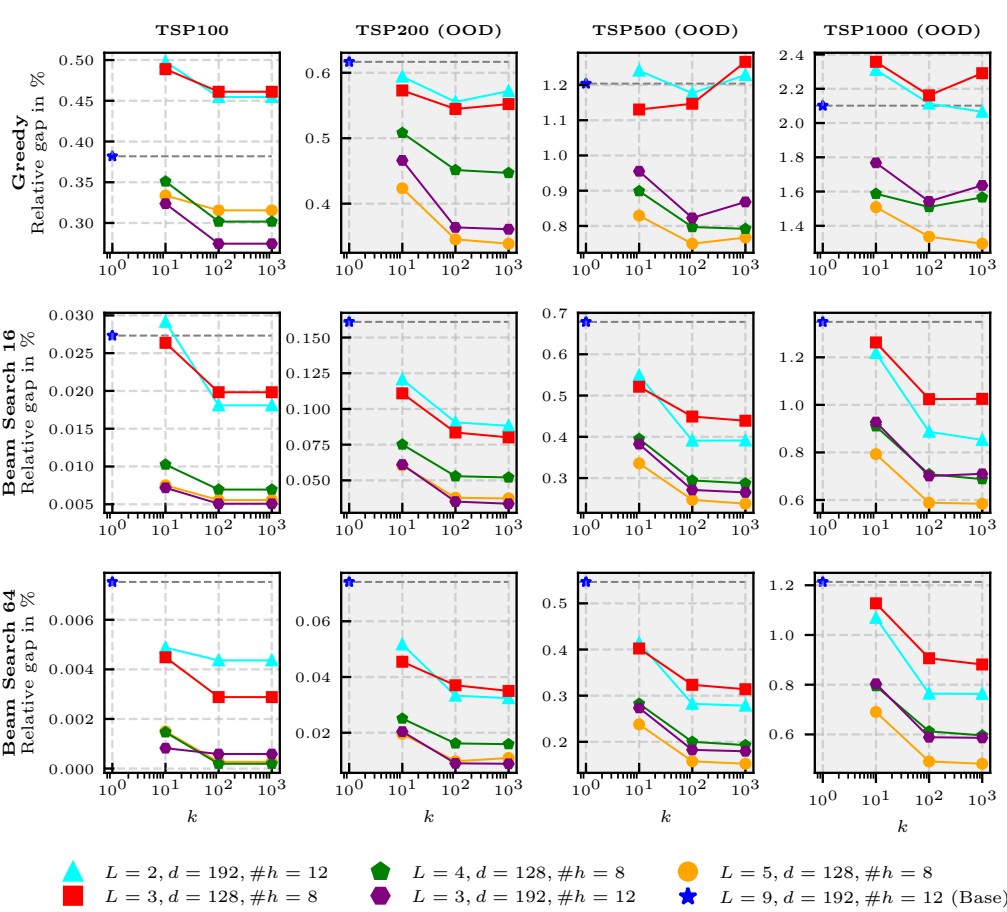

Figure 5: Additional results for the TSP. Recurrent models of different sizes are compared against their base encoder on the TSP with various TSP sizes up to 1000. All models are the same as in the main paper (see Fig 1). We show the relative gap of the models vs the number of steps the recurrent encoder is used in a row until the base encoder recomputes the embeddings ($k$). For solution times, see figure 4. All models were trained on the same imitation learning dataset of 1 million trajectories with problems of size 100. The structure of the model is given by $L$, the number of layers, $d$ the embedding dimension and $\#h$ the number of heads in the MHA mechanism.

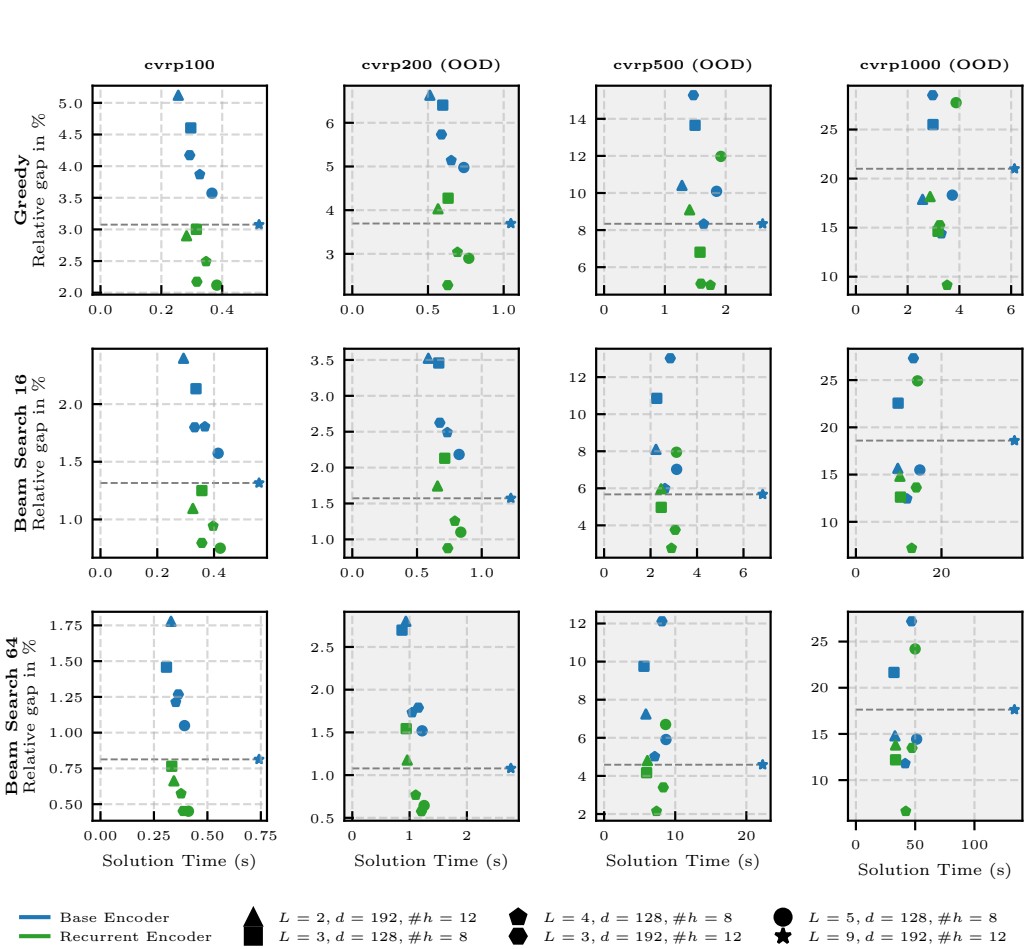

Figure 6: Additional results for recurrent models of different sizes vs base models of different sizes on the CVRP. All models are the same as in the main paper (see Fig 1). We show the relative gap of the models vs the time it takes to decode a single instance of the problem on ood instances of up to size 1000. All models use maximum $k = 1000$. For results with varying $k$, see figure 7. All models were trained on the same imitation learning dataset of 1 million trajectories with problems of size 100. The models in blue are differently sized configurations of non-recurrent models, while the models in green are recurrent with the recurrent encoder having the respective same size and structure where $L$, is the number of layers, $d$ is the embedding dimension and $\#h$ the number of heads in the MHA mechanism. All recurrent models always use the largest available base encoder and are trained with $k = 10$.

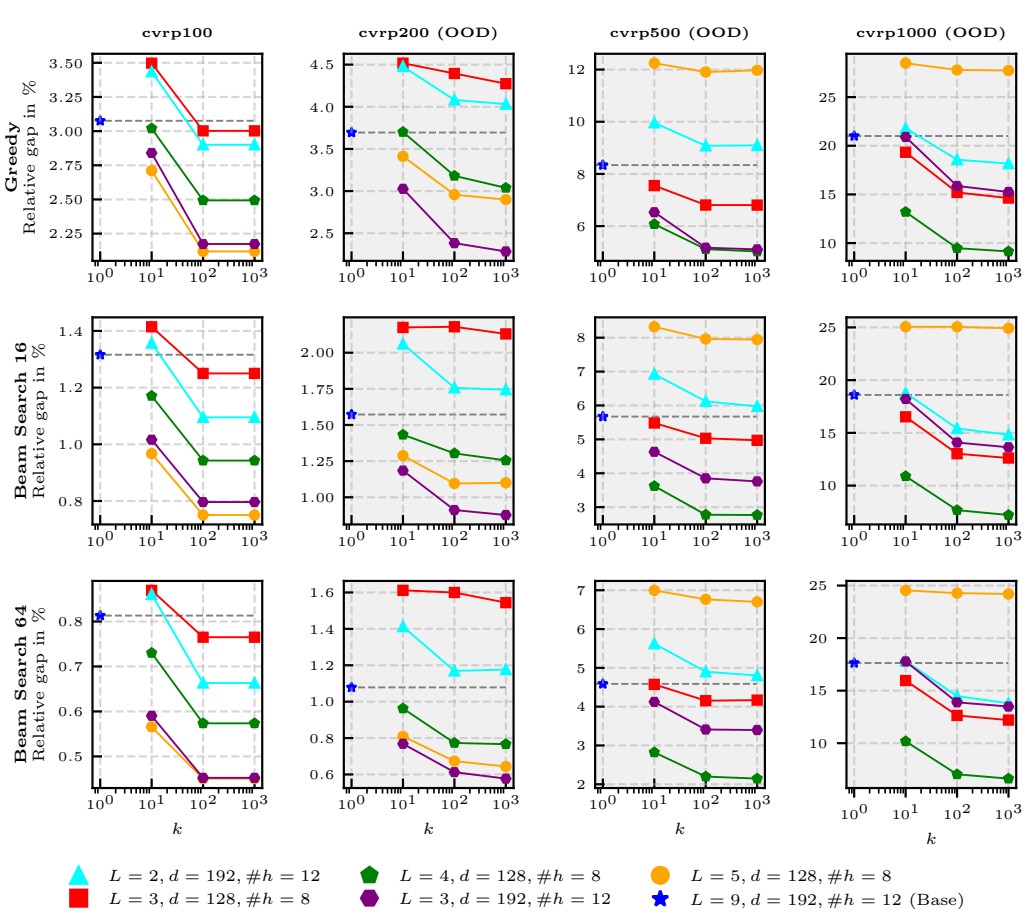

Figure 7: Additional results for the CVRP. Recurrent models of different sizes are compared against their base encoder on the CVRP with various CVRP sizes up to 1000. All models are the same as in the main paper (see Fig 1). We show the relative gap of the models vs the number of steps the recurrent encoder is used in a row until the base encoder recomputes the embeddings ($k$). For solution times, see figure 6. All models were trained on the same imitation learning dataset of 1 million trajectories with problems of size 100. The structure of the model is given by $L$, the number of layers, $d$ the embedding dimension and $\#h$ the number of heads in the MHA mechanism.

