# OpenReview forum: "Recurrent State Encoders for Efficient Neural Combinatorial Optimization"
_ICLR.cc/2026/Conference — Submitted to ICLR 2026_

### Official Review · Reviewer_dWHy · 2025-10-31

**Soundness:** 2
**Presentation:** 2
**Contribution:** 2
**Rating:** 2
**Confidence:** 3

**Summary:**

This paper proposes a neural network architecture for combinatorial optimization. Its key contribution is a recurrent module that generates node embeddings not only from the current state but also from the embeddings produced in previous steps. The neural solver is trained to imitate the behavior of a conventional solver used to generate the training data. The authors further integrate the trained network into a neighborhood search metaheuristic, using it as an oracle to generate initial solutions and iteratively improve them. Computational results show that this integration achieves comparable or better solution quality with lower computation time on TSP and CVRP instances, although performance lags behind baseline methods on OP instances.

**Strengths:**

- Overall, the paper is well written, and the intuition for designing the recurrent module is clearly articulated.

- The computational study is also comprehensive, covering well-known combinatorial problems that are commonly used in related literature.

**Weaknesses:**

- **Absolute performance margin**. My main concern lies in the practical significance of the reported improvements over baseline methods. The authors claim a (relative) 1.8–2.8× speed-up in inference, which is impressive. However, as shown in Figure 1, the inference times for existing neural combinatorial solvers are already under one second for TSP and CVRP instances with up to 200 nodes. In most applications, such problems are not solved in real time, and even when they are, current methods appear sufficiently fast. The small absolute performance gain in latency therefore raises questions about the broader motivation for studying this problem.

- **Imitation learning instead of reinforcement learning.** In my opinion, the primary goal of a neural combinatorial optimization solver is to produce high-quality solutions rather than simply achieving faster inference, since neural approaches are already much faster than traditional optimization algorithms. To ensure strong solution quality, the mainstream approach in recent literature has been to use reinforcement learning, whereas imitation learning is inherently limited by the performance of the solver used to generate the training data (as the authors also acknowledge). The reliance on imitation learning in the current implementation makes the y-axis of all figures difficult to interpret. I recommend that the authors (1) evaluate the proposed architecture using reinforcement learning–based training methods, and (2) report the y-axis as the relative optimality gap to facilitate a clearer understanding of the solver’s performance.

- **Performance.** In Figure 1, the proposed method performs worse than the baseline on OP instances, which raises concerns about the robustness of the approach. I find the idea of using the trained model as an oracle within a large neighborhood search scheme interesting. However, the y-axis in Figure 2 is difficult to interpret, as it appears to be reported relative to a potentially weak baseline, and the observed improvement margins are quite small (for example, from 0.3\% to 0.1\%)

**Questions:**

- **Section 3.1.** I found the description regarding the path-TSP formulation in Section 3.1 a bit confusing. In particular, how do you reformulate an TSP as a sequence of shortest path problems?

---

> ### Author Response · Authors · 2025-11-26
> **Rebuttal**
>
> We thank all reviewers for their time and feedback.
> ## Regarding Q1
> The path-TSP is a slight generalization of the TSP. In the TSP, we must find the shortest path that goes through all nodes exactly once and forms a cycle. In the path TSP we must find the shortest path that goes through all nodes exactly once but with a specified start and end node. Any TSP can be formulated as a path TSP by picking any node, duplicating it, and designating one of the duplicate nodes as the starting point and the other as the end point. Then the shortest path for the path TSP also forms the shortest cycle in the original TSP. This is just a simple trick that keeps the structure of the state representation the same for all states, since any subsequent state in the construction process also forms a valid path TSP instance. Any prior visited node gets removed and the current location becomes the new starting node.
> ## Regarding W1
> We agree that high solution quality is the primary goal for both combinatorial optimization and neural combinatorial  optimization therefore as well. However it is a mistake to ignore the trade-off between model precision and latency as not important. For any real application, the model will always be part of a larger search mechanism, where there exists a trade-off between performance and latency. If a model is very slow, less solutions given a fixed time budget can be explored. Thus, only prioritizing the models precision without the latency trade-off is a mistake. This is one of the reasons we encourage the literature to detail the whole accuracy runtime trade-off, as we have done in all our experiments. Otherwise, comparing models and methods at vastly different runtime points can be difficult. Our paper provides a principled approach to make the models more effective in terms of the precision latency trade-off.  We demonstrate this on very fundamental and widely tested combinatorial tasks and show that this significantly further benefits the overall search performance even under larger time budgets. This can be seen for example in our LNS, where for some problems we run the search for up to 200 seconds per instance (Figure 2,3).
> ## Regarding W2
> Since we investigate a principled change in the architecture and decoding mechanism, we have relied on the training setup as done in related prior work [3,2]. We have discussed possible alternatives for training in the paper and given the evidence in other mentioned work [1], there is no reason to believe that it could not work with other training mechanism. However the demonstration of this we leave for future work given the additional effort in implementation and computational expense.
> ## Regarding W3
> Crucially, we want to clarify that for the Orienteering Problem, our recurrent models only slightly drop in performance compared to the base encoder model in the out of distribution settings. In the in distribution setting, the models perform as well or better for all problem types. The main goal of NCO is to learn a solver for a distribution of instances. While some level of robustness to ood instances can be practically desirable, we do show this robustness in the TSP and CVRP. Thus we do not believe this to be a significant issue.
>
> We disagree that we measure performance against potentially weak baselines. All relative gaps are reported against the solvers outlined in section 4.1. Regarding figure 2, for the TSP this is Concorde, which produces optimal solutions and for the CVRP this is PyVRP which is a state of the art open source routing solver from the operations research community frequently used in the literature.
>
> [1] J. Pirnay and D. G. Grimm, “Self-Improvement for Neural Combinatorial Optimization: Sample Without Replacement, but Improvement,” _Transactions on Machine Learning Research_, Mar. 2024.
>
> [2] F. Luo, X. Lin, F. Liu, Q. Zhang, and Z. Wang, “Neural Combinatorial Optimization with Heavy Decoder: Toward Large Scale Generalization,” presented at the Thirty-seventh Conference on Neural Information Processing Systems, Nov. 2023.
>
> [3] D. Drakulic, S. Michel, F. Mai, A. Sors, and J.-M. Andreoli, “BQ-NCO: Bisimulation Quotienting for Efficient Neural Combinatorial Optimization,” presented at the Thirty-seventh Conference on Neural Information Processing Systems, Nov. 2023.

---

> > ### Comment · Reviewer_dWHy · 2025-11-26
> >
> > I thank the authors for their response.
> >
> > Regarding the path TSP reformulation, I now understand that this is a reformulation technique used to preserve the problem structure. However, I would avoid using the term “shortest path” because it holds a different meaning in the literature. A shortest path from A to B does not necessarily visit all nodes in the graph, yet my understanding is that this is required in your setting.
> >
> > I still have reservations about the significance of the contribution given the small absolute performance gains and the lack of convincing practical motivation for further accelerating neural CO solvers. I will therefore keep my rating.

---

> > > ### Author Response · Authors · 2025-11-27
> > >
> > > We thank the reviewer for their quick reply.
> > >
> > > We had hoped that under the context of "path-TSP" this would be clear but we understand that the formulation can lead to misunderstandings and will update it to be more clear that the requirement of visiting all nodes exactly once is maintained.
> > >
> > > While we respect the reviewers decision to not raise their score we want to make clear that we maintain our point of view. The trade-off between how much compute a heuristic takes (how fast it is) and how accurate it is, i.e. how likely its decision leads to optimality is absolutely fundamental to combinatorial optimization, since faster methods allow to search through more solutions at a fixed budget. This is true for any more involved search procedure regardless of it being a tree search, large neighborhood search or some other metaheuristic. Even in our experiments with LNS, we use long timeouts of up to 200 seconds. Even when only using a beam search, the required time can quickly become relevant. Our evaluations on TSP1000 in Figure 4 for example show that the base model requires ~2 minutes for a beam width of 64, searching only 64 different solutions. Our smallest recurrent model only takes ~30 seconds and does not drop in solution quality. We consider this very much to be relevant.

---

### Official Review · Reviewer_fUNw · 2025-10-31

**Soundness:** 2
**Presentation:** 2
**Contribution:** 2
**Rating:** 4
**Confidence:** 3

**Summary:**

This paper addresses inefficiencies in Neural Combinatorial Optimization (NCO), where traditional construction-based methods suffer from redundant computations—they fully recalculate state embeddings at each step despite minimal state changes (only one node is typically removed when adding a solution component). To solve this, this paper propose a recurrent encoder that enables lightweight updates: it reuses embeddings from the previous step, only adjusting for small state differences instead of recomputing from scratch.
Experiments on the Traveling Salesman Problem (TSP), Capacitated Vehicle Routing Problem (CVRP), and Orienteering Problem (OP) show the recurrent encoder outperforms or matches non-recurrent encoders (with 3× fewer layers) in solution quality, while cutting latency by 1.8–4×. It also exhibits strong robustness and practical value—integrating it into a Large Neighborhood Search (LNS) algorithm improves both efficiency and solution quality for large-scale problems.

**Strengths:**

I think this paper, by combining the base encoder and the recurrent encoder, attempts to solve the problem of redundant computations in the model training and inference process, which is interesting and has novelty.
1. Proposes a recurrent encoder that reuses prior step computations and only updates for small state changes, significantly reducing latency (1.8–4× faster) compared to non-recurrent encoders.
2. Demonstrates strong robustness—maintains stable performance even when the key hyperparameter k (recurrent update frequency) is much larger than the trained value.
3. Achieves comparable or better solution quality than non-recurrent counterparts while using 3× fewer layers, enabling model lightweighting.
4. Proves practical value by integrating with the Large Neighborhood Search (LNS) algorithm, improving both efficiency and solution quality.

**Weaknesses:**

1. Lacks a diagram that depicts the entire system architecture, which would help readers quickly grasp the key innovations.
2. Relies on imitation learning trained with expert trajectories, which depend on high-quality solutions from classical solvers and lack the ability to autonomously explore better strategies like reinforcement learning.
3. Requires periodic calibration via the base encoder (controlled by hyperparameter k), introducing additional tuning overhead.
4. Has not been tested on a wider range of combinatorial optimization problems (e.g., BPP, PSSP), making its generalizability across different problem types unproven.
5. The elements contained in Figures 1 and 2 in this paper are too numerous, and it takes a long time to understand them.
6. Cann't download opensource files from https://anonymous.4open.science/r/2CB0.

**Questions:**

In practical applications, the accuracy of the algorithm is undoubtedly the top priority. If the accuracy deteriorates, the practicality will be greatly affected. The paper mentions that using a recurrent encoder will accumulate errors, so it is necessary to periodically use the base encoder to recalculate. Moreover, the time and memory consumption of these baseline models compared in the paper are acceptable, so I think the motivation for accelerating with the recurrent encoder is not very solid.
Maybe the author could attempt to theoretically explain the benefits of using the concurrent encoder for the final model performance, rather than merely from the perspective of acceleration.
And, although in the experimental results, the methods presented in this paper outperform the baseline methods in some results, but the comparison is not very fair and incomplete (for example, POMO did not use reinforcement learning, only comparing scenarios with a problem size of 100 for training and other sizes for testing, and the test dataset did not use popular datasets such as TSPlib and CVRPlib).

---

> ### Author Response · Authors · 2025-11-26
> **Rebuttal**
>
> We thank all reviewers for their time and feedback.
> ### Regarding Q1,W4
> It is correct that in practical algorithms the accuracy is a top priority. However, for any real application, the model will always be part of a larger search mechanism, where there exists a trade-off between performance and latency. If a model is very slow, less solutions given a fixed time budget can be explored. Thus, only prioritizing the models precision without the latency trade-off is a mistake. This is one of the reasons we encourage the literature to detail the whole accuracy runtime trade-off, as we have done in all our experiments. Otherwise, comparing models and methods at vastly different runtime points can be difficult. Our paper provides a principled approach to make the models more effective in terms of the precision latency trade-off.  We demonstrate this on very fundamental and widely tested combinatorial tasks and show that this significantly further benefits the overall search performance even under larger time budgets. This can be seen for example in our LNS, where for some problems we run the search for up to 200 seconds per instance (Figure 2,3). While we need to restrict our experimental protocol to a few fundamental tasks, we are confident, that given the similarity in the construction processes in many tasks, benefits can also be found for other relevant problems. While there also will exist problems where the method might not lead to significant improvements, no one algorithm can be expected to be the perfect solution for all types of problems, given the difficult combinatorial nature of the tasks.
> ### Regarding W2
> Since we investigate a principled change in the architecture and decoding mechanism, we have relied on the training setup as done in related prior work [3,2]. We have discussed possible alternatives for training in the paper and given the evidence in other mentioned work [1], there is no reason to believe that it could not work with other training mechanisms.  However, the demonstration of this we leave for future work given the additional effort in implementation and computational expense with no great impact to our major findings.
> ### Regarding Q3
> We refute the notion that our comparisons are not fair. We are unsure what is meant by "POMO did not use Reinforcement Learning", since it is trained via Reinforcement Learning. Our method is also only trained on 100 sized problems and larger problems are ood as for many other methods. We did not include Pomo and some other models in the direct comparisons in Table 1 in Appendix F, since it is already outperformed by other baselines. We can also see that our raw model outputs without LNS outperform Pomo in any of the tested sizes and problems since in Figure 3, our line always starts below the Pomo performance, thus also the initial solution is already better than the one constructed by Pomo.
> ### Regarding W6
> We cannot replicate any problems downloading the source code from the provided link. Maybe there was a temporary outage of the anonymization service? Can you please check again. If it still does not work we can provide the source code through other means.
>
> [1] J. Pirnay and D. G. Grimm, “Self-Improvement for Neural Combinatorial Optimization: Sample Without Replacement, but Improvement,” _Transactions on Machine Learning Research_, Mar. 2024.
>
> [2] F. Luo, X. Lin, F. Liu, Q. Zhang, and Z. Wang, “Neural Combinatorial Optimization with Heavy Decoder: Toward Large Scale Generalization,” presented at the Thirty-seventh Conference on Neural Information Processing Systems, Nov. 2023.
>
> [3] D. Drakulic, S. Michel, F. Mai, A. Sors, and J.-M. Andreoli, “BQ-NCO: Bisimulation Quotienting for Efficient Neural Combinatorial Optimization,” presented at the Thirty-seventh Conference on Neural Information Processing Systems, Nov. 2023.

---

> > ### Comment · Reviewer_fUNw · 2025-11-26
> >
> > Thanks for your reply.
> > 1. The link is ok now, but last time, I was able to access this link, but there was no downloadable content within it.
> > 2. Our method is also only trained on 100 sized problems and larger problems are ood as for many other methods.—— Yes, but have you trained the models on instances across multiple sizes or all sizes and subsequent testing on instances of these sizes, if the method presented in the paper is also superior to other baseline methods on this kind of in-distribution scenario?

---

> > > ### Author Response · Authors · 2025-11-27
> > > **Training problems and link**
> > >
> > > Thank you for the quick reply.
> > >
> > > 1. Thank you for confirming that you can access the link now. There must have been some problem with the underlying service when you tried to access it last time, since we have not performed any modifications since before the submission deadline. You can see this also in the bottom left corner when opening the link where it states "Last Update: Sep 22, 2025".
> > >
> > > 2. Maybe there is still a misunderstanding about the training protocol. All training datasets exclusively consist of instances with exactly 100 customers. We do not train the models on any instances generated with more or less customers. Any evaluations performed on problems with more than 100 customers are out of distribution.

---

### Official Review · Reviewer_uugy · 2025-10-31

**Soundness:** 3
**Presentation:** 3
**Contribution:** 2
**Rating:** 4
**Confidence:** 4

**Summary:**

The paper proposes Recurrent State Encoders (RSE) for Neural Combinatorial Optimization (NCO), aiming to improve inference efficiency by leveraging temporal similarity between consecutive partial solutions.Instead of re-encoding the entire problem state at every construction step, RSE incrementally updates node embeddings using a recurrent encoder that combines the previous embeddings with the new partial solution features. A hyperparameter k controls how frequently the model performs full re-encoding.The method is evaluated on TSP, CVRP, and Orienteering Problem (OP) tasks, showing similar or slightly improved solution quality compared to baseline NCO models, while achieving up to 4× faster inference and maintaining robustness when integrated into a Large Neighborhood Search(LNS) framework.

**Strengths:**

1. The proposal to incrementally update embeddings using recurrent state memory is conceptually sound and well aligned with practical latency constraints in NCO applications.
2. Experiments span three canonical COPs (TSP, CVRP, OP) and multiple scales up to 1,000 nodes, comparing against a wide range of competitive baselines (e.g., BQ-NCO, LEHD, GLOP, POMO).

**Weaknesses:**

1. The proposed method is essentially an application of RNN-style computation reuse to NCO encoders, offering incremental rather than conceptual innovation.
2. No discussion on embedding stability, convergence, or representational drift during recurrent updates.
3. Models are trained solely via imitation learning; the effect of RL or self-improvement paradigms is not explored experimentally.
4. The paper reports performance improvements but offers little analysis of the underlying mechanisms driving them.

**Questions:**

1. How do the authors ensure that the recurrently updated embeddings remain stable and do not accumulate drift or noise over long decoding horizons, especially for large k or OOD instances?
2. Why are self-improvement models (e.g., Luo et al. 2024, Pirnay & Grimm 2024b) not compared directly, given that they represent the current state of the art in NCO efficiency?
3. Compared to a completely re-encoded transformer, does the representational power of recursive updates decrease? Is there any theoretical or experimental verification of the representational equivalence between the two?

---

> ### Author Response · Authors · 2025-11-26
> **Rebuttal**
>
> We thank all reviewers for their time and feedback.
> ### Regarding W3
> The focus of our paper is an investigation of a principled change in the architecture and decoding mechanism, in order to reuse computation done in prior states of the trajectory. As such we have relied on the training setup as done in related prior work [3,2], since the training paradigm is not part of our contributions. We have discussed possible alternatives for training in the paper and given the evidence in other mentioned work [1], there is no reason to believe that it could not work with other training mechanisms. However, the demonstration of this we leave for future work given the additional effort in implementation and computational expense with no great impact to our major findings.
> ### Regarding W2, Q1
> There are no ground truth correct embeddings, from which to measure a drift against, since the recurrent model is also trained directly against the best solution and not the base encoders embeddings. We argue the best measure for the stability of the recurrent updates is simply the solution quality that the model provides when executed with a recurrent update. Our experiments show clearly that when the recurrent models are executed within their training distribution and with a k smaller or equal than during training they always perform well even despite the possible drift that can occur due to the imitation learning and 2 stage training approach (ideally we would train jointly). Furthermore, even under heavy ood scenarios like 10 times the node size and a k 100 times larger than during training, the model still can show good performance, e.g. on the TSP, less than 1% gap to Concorde (Figure 4). While this can be surprising since we do not take any explicit precautions to ensure this stability and typically deep learning models can deteriorate in ood settings, it is our belief that this also makes the finding interesting for the broader NCO community.
>
> ### Regarding Q2:
> To the best of our knowledge, we cannot find any source code for Luo et al. 2024. This makes integration into our experimental protocol very difficult. Besides, as far as we can tell from the reported TSP1000 results, where their best reported value is 0.81% gap to LKH, our LNS quickly outperforms this (see figure 3). Our best recurrent models even outperform this only with beam search and no additional LNS (see figure 4).
>
> In Pirnay & Grimm 2024b their main contribution is an enhanced tree search procedure which then also gets used in a self improvement learning cycle. Both aspects are clearly orthogonal to our work. An improved tree search procedure would directly replace the used beam search for decoding, also within the LNS and would benefit from the speedup in the same way as now. And we have already discussed that the self-improvement style training is an interesting target also for our model but was not our main concern since we mainly wanted to showcase that it is indeed possible to speedup the inference without a significant sacrifice in model accuracy through the recurrent style of computation. As such we consider further enhancements in training and tree search to be future work and don't believe that further comparisons to the mentioned work provide significant additional insights that impact the main findings of our paper.
>
> ### Regarding W4, Q3
> Since the recurrent model does have full access to the current state **and** prior embeddings, it could in principle learn to fully ignore the prior embeddings and only compute from the current state. In that case, if both the recurrent and base encoder have the same number of layers, they could learn an equivalent model with the same representational power. As soon as useful information for the next step can be passed on in the form of the prior embeddings, we can reduce the amount of computation in the recurrent model. This is also what we show what actually happens in practice in our problems, since even cutting the number of layers to a third of the base encoder, the original performance can be roughly matched.

---

### Meta-Review · Area_Chair_8rY3 · 2025-12-28

**Summary:**

This paper introduces a recurrent encoder architecture for neural combinatorial optimization (NCO), aiming to reduce redundant computation by reusing prior state embeddings during solution construction. The method is evaluated on standard benchmarks (TSP, CVRP, OP), demonstrating 1.8–4× faster inference with similar or slightly improved solution quality compared to standard (non-recurrent) encoders, and competitive performance when integrated with a large neighborhood search.

While the paper presents a useful efficiency improvement with solid experiments and clear writing, the contribution is largely incremental, and practical benefits seem limited under realistic conditions, as existing NCO models are already fast for moderate-sized problems (especially linear transformers in Luo et.al.2025 with the same O(n^2) time complexity). The current form is clear below the acceptance threshold.

**Reviewer Concerns:**

The main concern from reviewers is about the main innovation—using recurrent updates for embeddings—is incremental and conceptually straightforward in deep learning, leading several reviewers to view the contribution as rather limited.

**Reviewer Scores:**

Reviewers provided clear evidence for their insightful concerns.

---

### Decision · Program_Chairs · 2026-01-26

Reject